# *Chryseobacterium herbae* Isolated from the Rhizospheric Soil of *Pyrola calliantha H. Andres* in Segrila Mountain on the Tibetan Plateau

**DOI:** 10.3390/microorganisms11082017

**Published:** 2023-08-05

**Authors:** Li Zhang, Yan Wang, Delong Kong, Qingyun Ma, Yan Li, Zhen Xing, Zhiyong Ruan

**Affiliations:** 1College of Life Sciences, Yantai University, Yantai 264005, China; wkiaazl@outlook.com; 2CAAS-CIAT Joint Laboratory in Advanced Technologies for Sustainable Agriculture, Institute of Agricultural Resources and Regional Planning, Chinese Academy of Agricultural Sciences, Beijing 100081, China; a18301335070@163.com (Y.W.); kongdelong009@163.com (D.K.); mqy@webmail.hzau.edu.cn (Q.M.); 3College of Resources and Environment, Tibet Agricultural and Animal Husbandry University, Linzhi 860000, China; xztibetan@163.com; 4State Key Laboratory of Agricultural Microbiology, Huazhong Agricultural University, Wuhan 430070, China

**Keywords:** *Chryseobacterium herbae* sp. nov., alpine ecosystems, rhizosphere soil, Gram-staining-negative, *Pyrola calliantha H. Andres*, genomic, shotgun proteomics

## Abstract

A non-motile, Gram-staining-negative, orange-pigmented bacterium called *herbae* pc1-10^T^ was discovered in Tibet in the soil around *Pyrola calliantha H. Andres*’ roots. The isolate thrived in the temperature range of 10–30 °C (optimal, 25 °C), pH range of 5.0–9.0 (optimum, pH = 6.0), and the NaCl concentration range of 0–1.8% (optimal, 0%). The DNA G+C content of the novel strain was 37.94 mol%. It showed the function of dissolving organophosphorus, acquiring iron from the environment by siderophore and producing indole acetic acid. Moreover, the genome of strain *herbae* pc1-10^T^ harbors two antibiotic resistance genes (IND-4 and AdeF) encoding a β-lactamase, and the membrane fusion protein of the multidrug efflux complex AdeFGH; antibiotic-resistance-related proteins were detected using the Shotgun proteomics technology. The OrthoANIu values between strains *Chryseobacterium herbae* pc1-10^T^; *Chryseobacterium oleae CT348^T^*; *Chryseobacterium kwangjuense KJ1R5*^T^; and *Chryseobacterium vrystaatense R-23566^T^* were 90.94%, 82.96%, and 85.19%, respectively. The in silico DDH values between strains *herbae* pc1-10^T^; *C. oleae* CT348^T^; C. *kwangjuense* KJ1R5^T^; and *C. vrystaatense* R-23566^T^ were 41.7%, 26.6%, and 29.7%, respectively. *Chryseobacterium oleae*, *Chryseobacterium vrystaatense*, and *Chryseobacterium kwangjuense*, which had 16S rRNA gene sequence similarity scores of 97.80%, 97.52%, and 96.75%, respectively, were its closest phylogenetic relatives. *Chryseobacterium herbae* sp. nov. is proposed as the designation for the strain *herbae* pc1-10^T^ (=GDMCC 1.3255 = JCM 35711), which represented a type species based on genotypic and morphological characteristics. This study provides deep knowledge of a *Chryseobacterium herbae* characteristic description and urges the need for further genomic studies on microorganisms living in alpine ecosystems, especially around medicinal plants.

## 1. Introduction

The genus *Chryseobacterium* is included within the family *Weeksellaceae* [1], the main bacterial lineage in the phylum Bacteroidetes. It is well known that strains of the plant-related *Chryseobacterium* species have the ability to promote plant growth [2,3,4,5,6] and degrade pesticide residues [7,8], and many of them have genes associated with antibiotic resistance [6,9,10,11,12]. According to the “List of Prokaryotic Names with Standing in Nomenclature” (http://www.bacterio.net/ accessed on 20 May 2023) which interprets the list of all validly published names within this genus, 169 species with validly published names were at the time of writing considered to be members of the genus *Chryseobacterium*. Animals [13,14,15,16], plants [17,18,19,20,21,22], water [23,24,25], sediment [23,26,27], and air [28] are just a few of the diverse settings in which members of the genus *Chryseobacterium* can be isolated.

In the southeast of the Tibetan Plateau, where alpine forest ecosystems play a significant role in maintaining the water balance and carbon sequestration, there are abundant forestry resources to be discovered [29]. Soil microbes play a vital role in biogeochemical cycling processes in alpine forest ecosystems [30,31,32]. In China, *Pyrola calliantha* is found in many provinces, including Shaanxi, Qinghai, Gansu, and Henan. *P. calliantha* often inhabits broad-leaved woods, coniferous and broad-leaved mixed forests, or mountain coniferous forests between 700 and 4100 m above sea level. *Pyrola calliantha* has been used medicinally for a very long time in China. It is a traditional Chinese medication that, after drying, has a wide range of therapeutic characteristics. It is frequently used to cure disorders including bleeding, coughing up blood, inflamed sores, and discomfort in tendons and bones [33]. 

This study aims to characterize a new species, *Chryseobacterium herbae* pc1-10^T^, which inhabits the rhizosphere of *Pyrola calliantha*, with genomics and proteomics-assisted characterization.

## 2. Materials and Methods

### 2.1. Bacterial Isolation and Cultivation

The strain of bacterium *Chryseobacterium* sp. pc1-10^T^. was isolated from the root system soil of *Pyrola rotundifolia* (*Pyrola calliantha H. Andres.*) in Segrila Mountain, geographically located in Nyingchi District, Tibet Autonomous Region, China (94°34′34.67″ E, 29°33′41.85″ N). The samples were kept in the lab at 4 °C until use and transported at low temperatures. In a sterile mineral buffer, the root system soil samples were suspended and homogenized [34]. On Nutrient Agar (NA; Difco, Beirut, Lebanon) adjusted to pH 7.0, serial dilutions were inoculated and incubated for 5 days at 30 °C. A total of 29 morphologically distinct colonies appeared on nutrient agar plates. We obtained 15 strains, strain *herbae* pc1-10^T^ being one of them. The pure culture was then kept in glycerol suspensions (25%, *v*/*v*) at −80 °C and on NA slants at 4 °C.

### 2.2. Phylogenetic and Genotypic Analysis

Using bacterial universal primers 27F (5′-AGAGTTTGATCCTGGCTCAG-3′) and 1492R (5′-GGTTACCTTGTTACGACTT- 3′), the 16S rRNA gene of the *herbae* pc1-10^T^ was amplified by PCR [35]. The Life Technologies Company (Shenzhen, China) sequenced purified PCR products (approximately 1.4 kb). The virtually complete sequences of 16S rRNA genes were put together using the DNAMan version 6.0.3.99 software (DNASTAR Inc., Madison, WI, USA). All of the 16S rRNA gene sequences of the closest phylogenetic members were used to construct phylogenetic trees using neighbor-joining (NJ), maximum-likelihood (ML), and silva alignment methods by the Mega 7 software [36]. 

### 2.3. Physiological, Biochemical, and Chemotaxonomic Analysis

An exponentially developing culture’s cell morphology was investigated using a Nikon 80i light microscope and a transmission electron microscope (Hitachi 7500, Tokyo, Japan). The isolates were grown on Nutrient Agar (NA; Difco) in a variety of environments to assess physiological traits. Five temperatures of 10, 20, 25, 30, and 40 °C were set to determine the growth temperature. The NaCl-free NA broth (Difco, made according to the NA recipe but without agar and NaCl) with various NaCl concentrations (0, 0.1–2.0% at 0.1% increments, and 2–12% at 1% increments, *w*/*v*) was used to investigate the optimum concentration of NaCl for growth. The hydrolysis of Tweens 20, 40, 60, and 80 was discovered using conventional techniques [37]. *Herbae* pc1-10^T^ was inoculated into an NA medium with a 0.05% L-tryptophan with the NA medium serving as a control in order to determine the production of indole acetic acid (IAA) [38]. The culture was incubated for 48 h at 25 °C. One milliliter of bacterial culture was centrifuged at 10,000 rpm for 6 min at 4 °C, and 0.6 mL of supernatant was mixed with 0.6 milliliters of Salkowski’s reagent (17.5 milliliters of HClO_4_, 32.5 milliliters of distilled water, 1 milliliter of 0.5 M FeCl_3_·6H_2_O) and the mixture was then incubated for thirty minutes at room temperature and in the dark [39]. At 530 nm, color absorbance was measured, and a quantitative analysis was performed by comparing the results to the IAA standard (5–40 mg/L). Chrome azurol S (CAS) agar medium was used to test *herbae* pc1-10^T^’s capacity to produce siderophores [40]. Briefly, in order to examine color change, a 5 μL inoculum of the *herbae* pc1-10^T^ was spot plated and stored at 25 °C. To test its nitrogen fixation capacity, we inoculated it on the ASHBY medium by observing if it grows. Additional physiological and biochemical features were determined as follows. 

Transmission electron microscopy (TEM) was used to analyze the strain *herbae* pc1-10^T^’s morphological characteristics, including the size and shape of the cells and their surface ornamentation. Gram staining was performed according to Halebian et al. [41]. Filter-paper disks (Hopebio, Qingdao, China) impregnated with a 1% solution of N,N-Dimethyl-p-phenylenediamine dihydrochloride were used to measure the activity of oxidase, a positive test resulted in the production of a blue-purple coating on the filter paper biomass within two minutes. In order to evaluate the effects of starch degradation, plates containing nutrient agar (NA Difco) (0.8%), starch (1%), and agarose (1.5%) were used. These plates were developed by flooding them with iodine solution (1%) after 5 days of incubation, and hydrolysis rings around the strain were then observed to form. According to the manufacturer’s (bioMérieux, Marcy-l’Étoile, France) instructions, API ZYM galleries were used to assess the strain *herbae* pc1-10^T^’s enzyme activity. Other biochemical assays were carried out in accordance with Tindall et al.’s instructions, including the methyl red (method 15.2.52) and the Voges–Proskauer (method 15.2.82) tests [42]. The utilization of carbon compounds was measured using API 20NE test strips (bioMérieux) according to the manufacturer’s instructions, incubated at 25 °C. The growth of bacteria on various carbohydrates and their derivatives (heterosides, polyalcohols, and uronic acids) was examined using the API 50CH test.

Well-grown cells of strain *herbae* pc1-10^T^ grown on tryptic soy agar (TSA) medium for 24 h at 25 °C were used for fatty acid analysis. Separation and identification of whole-cell esters were carried out with the Sherlock Microbial Identification System (MIDI version 3.0) as represented by Vandamme et al. [43]. Polar lipids were extracted using the method of Xu et al. [44] and separated by two-dimensional TLC using silica gel 60 F254 aluminum-backed thin-layer plates (Merck, Darmstadt, Germany) [45]. Isoprenoid quinones of strain *herbae* pc1-10^T^ were extracted from freeze-dried cells and analyzed as previously described using LC-MS [46]. Using a molybdophosphoric acid hydrate ethanol solution, the total lipids were identified. Utilizing the ninhydrin reagent, aminolipids were identified. With the help of the Zinzadze reagent, phospholipids were identified, and the naphthol reagent was used to find glycolipids. The data were interpreted as described by Tindall et al. [42].

### 2.4. Genome Analyses

Genomic DNA was collected according to the method of Marmur [47]. The genomic G+C content and the organism overview of strain *herbae* pc1-10^T^ were determined using the RAST Server [48] by genome sequence, which was performed on the Illumina MiSeq platform by the Guangzhou Magigene Company (Guangzhou, China). The reads were put together using the SOAPdenovo version 1 [49]. According to the minimal requirements suggested by Chun et al., the average nucleotide identity (ANI) and in silico DNA–DNA hybridization (DDH) values were determined [50]. The OrthoANIu algorithm (https://www.ezbiocloud.net/tools/ani accessed on 17 July 2022) was used to determine the ANI value between the two genomes [49]. The Genome-to-Genome Distance Calculator 3.0 (https://ggdc.dsmz.de/ggdc.php# accessed on 17 July 2022) was used to determine the in silico DDH values [51]. Under the accession number JAOAMU000000000, strain *herbae* pc1-10^T^’s Whole-Genome Shotgum project was deposited at GenBank. 

### 2.5. Shutgun Proteomics Analyses

We used shotgun proteomics [52,53,54] to detect the global profile of the protein/polypeptide complement within the mixture expression in the cell and culture supernatant of strain *herbae* pc1-10 ^T^. The cells and the supernatant of *herbae* pc1-10^T^ were separated by centrifugation (8000 rpm, 10 min). The cells were washed three times with a 0.85% NaCl, immediately frozen in liquid nitrogen, and transported to Personal Biotechnology Co., Ltd., Shanghai, China at −80 °C until use. The protein sequence database is QLDBPSN025. 

### 2.6. Pigment Analyses

We experimentally validate the production of flexirubin by *herbae* pc1-10^T^ as described by Siddaramappa et al. [55] by exposing a 2 mL pure culture of strain *herbae* pc1-10^T^, which was cultured in an NA liquid medium for 24 h, to a 500 μL 3% KOH, which resulted in a change from yellow to orange/red if flexirubin pigments were present, followed by a neutralization step with a 500 μL 1.5 N HCl which resulted in a return to yellow pigmentation. 

## 3. Results and Discussion

### 3.1. Phylogenetic and Genotypic Analysis

The 16S rRNA gene sequence (1517 bp) of strain *herbae* pc1-10^T^ (OP352779) was acquired. Comparative analysis from this 16S rRNA gene sequence showed that the novel strain stands for a member of the genus *Chryseobacterium*. Strains *Chryseobacterium oleae* CT348^T^, *Chryseobacterium vrystaatense* RJ-7-14^T^, and *Chryseobacterium kwangjuense* KJ1R5^T^, were prominently picked out as reference strains for comparative studies. Phylogenetic analysis grounded on the NJ, ML methods showed that strain *herbae* pc1-10^T^ formed a stable subclade with *C. oleae* CT348^T^ in the phylogenetic tree (Figure 1 and Appendix A).

### 3.2. Physiological, Biochemical, and Chemotaxonomic Analysis

Cells (0.9–2.4 µm long and 0.5–0.7 µm wide) were observed to be aerobic (Appendix A), Gram-staining-negative, rod-shaped, and non-motile. Colonies on nutrient agar were detected as orange-pigmented, entire, convex, and circular (Appendix A). On the NA agar, a colony with a size of 0.5–1 mm was seen for 5 days at 25 °C. At pH 5.0–9.0 and 10–30 °C (optimal: 25 °C), cells grew; pH 6.0 was optimum. Cells were found to grow most effectively in the absence of NaCl yet tolerated 1.8% NaCl. Catalase and oxidase, on the other hand, were determined to be negative. The strain *herbae* pc1-10^T^ significantly grew well and was observed to generate a halo of organophosphorus-dissolving circles but no degradant halo of phosphate on their respective media. The strain was cultured in an NA-tryptophan liquid medium for 2 days and an IAA yield was detected to be 3.53 mg/L (Appendix A, Appendix A). Unfortunately, the strain *herbae* pc1-10^T^ did not grow on the CAS medium (with nutrient agar providing the essential elements for growth) and the ASHBY medium. In the pigment detection experiment, after adding the KOH solution to the bacterial suspension, the color changes from yellow to orange–red, and the addition of HCl does not change color (Appendix A), which can prove its production of the flexirubin-type pigment.

Iso-C_15:0_ (42.82%), iso-C_17:0_ 3-OH (21.03%), and Summed features 9 (C_16:1_ w6c and/or C_16:1_ w7c; 12.43%) were the strain *herbae* pc1-10^T^’s main cellular fatty acids (>10%). The amount of summed feature 9 (C_16:1_ w6c and/or C_16:1_ w7c; 12.43%) in strain *herbae* pc1-10^T^ was lower than that tested in *C. oleae* CT348^T^; C. *kwangjuense* KJ1R5^T^; C. *vrystaatense* R-23566^T^. However, strain *herbae* pc1-10^T^ stood apart from other phylogenetically related members of the genus *Chryseobacterium* due to quantitative variations in major and minor fatty acids and the presence of specific minor fatty acids like C_16:0_ 3-OH, iso-C_15:0_ 3-OH, iso-C_16:0_, etc. (Table 1). Menaquinone-6 (MK-6) was found to be the only respiratory quinone that fit the description of the genus *Chryseobacterium*. Phosphatidylethanolamine (PE) contains the main polar lipid. Additionally, two lipids (L1-L2), three glycolipids, and five aminolipids (AL1-AL5) were found (Appendix A). 

Summed Features 3: including C_16:1_
*ω*6c and/or C_16:1_
*ω*7c, Summed Features 9: 10-methyl C_16:0_ and/or iso-C_17:1_ *ω*9c.

In the API 50CH strip, weak acid production from Erythritol, Glucose, Fructose, Mannose, Trehalose, and Gentiobiose, and acid production from Glycerol, d-Arabinose, L-Arabinose, D-Ribose, D-Xylose, L-Xylose, D-Adonitol, Methyl-β-D-xylopyranoside, D-Galactose, L-Sorbose, L-Rhamnose, Dulcitol, Inositol, Mannitol, Sorbitol, Methyl-α-D-mannopyranoside, Methyl-α-D-glucopyranoside, N-Acetyl-glucosamine, Amygdalin, Arbutin, Esculin, Salicin, Cellobiose, Maltose, Lactose, Melibiose, Sucrose, Inulin, Melezitose, Raffinose, Amyloid, Glycogen, Xylitol, D-Turanose, D-Lyxose, D-Tagatose, D-Fucose, L-Fucose, D-Arabitol, L-Arabitol, Glucoheptonate, 2-Ketogluconate, 5-Ketogluconate were negative. In the API ZYM strip, there were observations of positive results for Alkaline phosphatase, Esterase (C4), Esterase Lipase (C8), Lipase (C14), Leucine arylamidase, Acid phosphatase, Naphthol-AS-BI-phosphohydrolase, α-Galactosidase, β-Glucosidase, N-Acetyl-β-glucosaminidase, α-Fucosidase. However, other negative results for Valine arylamidase, Cystine arylamidase, Trypsin, Chymotrypsin, β-Galactosidase, β-Glucuronidase, α-Glucosidase, and α-Mannosidase were obtained. Strain *herbae* pc1-10^T^ showed negative results for Nitrate reduction, D-Glucose fermentation, Arginine dihydrolase, Urease, β-Galactosidase, Arabinose, N-Acetyl-D-glucosamine, D-Maltose, Gluconate, Capric acid, Adipic acid, Malic acid, Citrate, p-hydroxy-Phenylacetic acid and positive results for β-Glucosidase, Indole production, Gelatinase and weakly positive for Glucose, Mannitol, mannose in the API 20NE strip test. Table 2 compares strains from the genus *Chryseobacterium* that are phylogenetically linked to strain *herbae* pc1-10^T^ in terms of how well they thrive on various carbohydrates and their derivatives. 

### 3.3. Genome Analyses

The genome of strain *herbae* pc1-10^T^ was 5,142,603 bp long including 36 contigs with an N50 value of 306,357 and a genome coverage of 81.0×. The DNA G+C content was 37.94 mol%, the mean length was 142,836.5 bp, and 71 tRNA genes, 3 rRNA genes, and 1 tmRNA were predicted. The strain *herbae* pc1-10^T^ full-length 16S rRNA scaffold was retrieved from the genome assembly to confirm that the genomic data is legitimate and uncontaminated, and one 16S rRNA sequence was received. The result was consistent with that of the PCR sequence. The OrthoANIu values between strains *C. herbae* pc1-10^T^; *C. oleae* CT348^T^; C. *kwangjuense* KJ1R5^T^; and *C. vrystaatense* R-23566^T^ were 90.94%, 82.96%, and 85.19%, respectively. The in silico DDH values between strains *herbae* pc1-10^T^; *C. oleae* CT348^T^; C. *kwangjuense* KJ1R5^T^; and *C. vrystaatense* R-23566^T^ were 41.7%, 26.6%, and 29.7%, respectively.

The *herbae* pc1-10^T^ genome’s RAST annotation showed 264 subsystems with 4733 genes and an 18% subsystem coverage, totaling 264 subsystems (see Figure 2 below for details). It provides initial annotations of gene functions for *herbae* pc1-10^T^’s genomes, such as producing siderophores, plant hormones, beta-lactamase, etc.

Furthermore, we annotated the genome of *herbae* pc1-10^T^ through Proksee (https://proksee.ca/projects/new accessed on 21 May 2023) (Figure 2). Antibiotic resistance gene predictions showed that *herbae* pc1-10^T^ contained two antibiotic resistance genes: IND-4, which is a beta-lactamase found in *Chryseobacterium indologenes*, and AdeF, which is the membrane fusion protein of the multidrug efflux complex AdeFGH; the protein-coding information is provided by CARD [59] (https://card.mcmaster.ca/ accessed on 21 May 2023). According to a study, bacteria that live in soil or water naturally have ARGs that they can use to compete with other organisms that share the same habitat [60]. We used CRISPRCasFinder [61] to identify CRISPR (clustered regularly interspaced short palindromic repeats) arrays; 5 of the 23 analysis sequences were found to contain CRISPR but no Cas genes were found in the vicinity of CRISPRs. Components of these biological systems can be used in multiple applications in genetic engineering [62,63].

Secondary metabolites (Table 3) were identified using antiSMASH version 7.0.0 (https://antismash.secondarymetabolites.org/, accessed on 14 May 2023). One gene cluster with arylpolyene and resorcinol biosynthesis domains was found in strain *herbae* pc1-10^T^, and it showed a 75% similarity to a known biosynthetic gene cluster generating flexirubin, the major pigment of *Chryseobacterium* [64]. Strain *herbae* pc1-10^T^ additionally had a gene cluster with NI-siderophore biosynthetic domains (Figure 3d) showing a 33% similarity to fulvivirgamide A2 biosynthetic gene cluster from *Fulvivirga marina* [65] and lanthipeptide class I (Figure 3e) biosynthetic domains showing a 17% similarity to pinensins biosynthetic gene cluster from *Chitinophaga pinensis DSM 2588* [66], which is the first antifungal lantibiotics [67]. Other gene clusters containing microviridin (Figure 3b), hglE-KS, T1PKS, terpene (Figure 3c), and lanthipeptide class I (Figure 3a) biosynthetic were observed with no similarity to known clusters. The serine proteases chymotrypsin, trypsin, and elastase are just a few of the ones that microviridins are effective and selective inhibitors of [68]. Furthermore, it was discovered that microviridin J was poisonous to *Crustacean Daphnia* [69].

### 3.4. Shut-Gun Proteomics Results

The qualitative statistical results of proteins are shown in the attached Appendix A. Finally, we identified 2455 (cell) and 2650 (supernatant) proteins and 19,568 (cell) and 23,825 (supernatant) peptides from the two samples, respectively (see Appendix A Appendix A). 

The TonB-dependent siderophore receptor (MCT2563047.1), the biopolymer transporter ExbD (MCT2562352.1, MCT2562353.1, MCT2563473.1), and the MotA/TolQ/ExbB proton channel family protein (MCT2562351.1) were detected in both cell and supernatant, which means that *herbae* pc1-10^T^ could uptake siderophore from the outside environment [70]. The pinensin family lanthipeptide (MCT2564031.1) was detected in the supernatant, and it was regarded as the first lantibiotics isolated from a Gram-negative native producer [67]. It is famous for its significant resistance to some fungi; however, we found that *herbae* pc1-10^T^ only had a weak inhibitory effect on the growth of pathogenic fungi (*Magnaporthe oryzae*) of rice blast (Appendix A). Terpene synthase family protein (MCT2561927.1) was detected in the supernatant and cell. Penicillin-binding transpeptidase domain-containing protein (MCT2563304.1), the TolC family protein (MCT2563484.1, MCT2561713.1, MCT2564157.1), nitroreductase (MCT2564731.1) [71], and the efflux transporter outer membrane subunit (MCT2563723.1) were detected in cell samples. These proteins are closely related to antibiotic resistance. However, microviridin was not detected in the culture environment and cells, inferring that the predicted result is unreliable, or that the gene is not expressed in our culture environment, or the expression level does not reach the proteome detection threshold, etc.

## 4. Description of *Chryseobacterium herbae* sp. nov.

### Chryseobacterium herbae (her’bae. L. gen. n. herbae, of a herb)

Cells of a new strain *herbae* pc1-10^T^ are rod-shaped, aerobic, Gram-stain-negative, and non-motile, measuring 0.9 to 2.4 μm in length and 0.5 to 0.7 μm in width. Nutrient agar colonies are rounded, whole, convex, and orange-colored (Flexirubin-type pigment). The colony size on the NA agar is 0.5–1 mm for 5 days at 25 °C. The optimum temperature for cell growth is 25 °C, while the optimum pH range is 5.0 to 9.0. NaCl is not necessary for good cell growth, but a 1.8% NaCl is acceptable. Catalase and oxidase are positive and negative, respectively. Starch, Tweens 20, 40, 60, and 80 are hydrolyzed but arabinose or urea are not decomposed. It produces gelatinase and β-glucosidase but not β-galactosidase and arginine dihydrolase. Iso-C_15:0_, iso-C_17:0_ 3-OH, and summed features 9 (C_16:1_ ω6c and/or C_16:1_ ω7c) were the main cellular fatty acids (>10%) of strain *herbae* pc1-10^T^, while menaquinone- 6 (MK-6) was the predominate respiratory quinone. 

Weak acid production from Erythritol, Glucose, Fructose, Mannose, Trehalose, and Gentiobiose, and acid production from Glycerol, d-Arabinose, L-Arabinose, D-Ribose, D-Xylose, L-Xylose, D-Adonitol, Methyl-β-D-xylopyranoside, D-Galactose, L-Sorbose, L-Rhamnose, Dulcitol, Inositol, Mannitol, Sorbitol, Methyl-α-D-mannopyranoside, Methyl-α-D-glucopyranoside, N-Acetyl-glucosamine, Amygdalin, Arbutin, Esculin, Salicin, Cellobiose, Maltose, Lactose, Melibiose, Sucrose, Inulin, Melezitose, Raffinose, Amyloid, Glycogen, Xylitol, D-Turanose, D-Lyxose, D-Tagatose, D-Fucose, L-Fucose, D-Arabitol, L-Arabitol, Glucoheptonate, 2-Ketogluconate, 5-Ketogluconate was negative. In the API ZYM strip, there were observations of positive results for Alkaline phosphatase, Esterase (C4), Esterase Lipase (C8), Lipase (C14), Leucine arylamidase, Acid phosphatase, Naphthol-AS-BI-phosphohydrolase, α-Galactosidase, β-Glucosidase, N-Acetyl-β-glucosaminidase, α-Fucosidase. However, other negative results for Valine arylamidase, Cystine arylamidase, Trypsin, Chymotrypsin, β-Galactosidase, β-Glucuronidase, α-Glucosidase, and α-Mannosidase were obtained. Strain *herbae* pc1-10^T^ showed negative results for Nitrate reduction, D-Glucose fermentation, Arginine dihydrolase, Urease, β-Galactosidase, Arabinose, N-Acetyl-D-glucosamine, D-Maltose, Gluconate, Capric acid, Adipic acid, Malic acid, Citrate, p-hydroxy-Phenylacetic acid and positive results for β-Glucosidase, Indole production, Gelatinase and weakly positive for Glucose, Mannitol, mannose in the API 20NE strip test.

The type strain’s genomic DNA has a G+C concentration of 37.9 mol%. Genome sequence accession number: JAOAMU000000000. 16S rRNA gene accession number: OP352779. The type strain, *herbae* pc1-10^T^ (=GDMCC 1.3255^T^ = JCM 35711^T^), was isolated from the root system soil of *Pyrola calliantha H.* in Tibet. 

## 5. Discussion

Rhizosphere microbial colonization often has plant genotype specific selectivity [72]. With technological progress, genomics and proteomics provide more sufficient technical support for microbial molecular level research [54]. The strain *herbae* pc1-10^T^ initially showed the function of dissolving organophosphorus, siderophore-mediated iron acquisition systems, and producing indole acetic acid (3.53 mg/L in 2 days), which may have a positive effect on plant growth. Moreover, the genome of strain *herbae* pc1-10^T^ harbors two antibiotic resistance genes (IND-4 and AdeF) encoding a β-lactamase, which is responsible for β-lactam antibiotic resistance, and encoding the membrane fusion protein of the multidrug efflux complex AdeFGH, respectively. They were confirmed in Shut-gun proteomic testing. Due to the fact that bacteria that live naturally in soil or water have inbuilt ARGs to combat the chemical substances created by rivals living in the same environment, we speculate that the antibiotic resistance gene possessed by strain *herbae* pc1-10^T^ is caused by related chemicals present in the rhizosphere of *Pyrola calliantha H. Andres*. In a word, analyzing the internal characteristics of microorganisms in special habitats may provide new ideas for future production practices and scientific research.

## Figures and Tables

**Figure 1 microorganisms-11-02017-f001:**
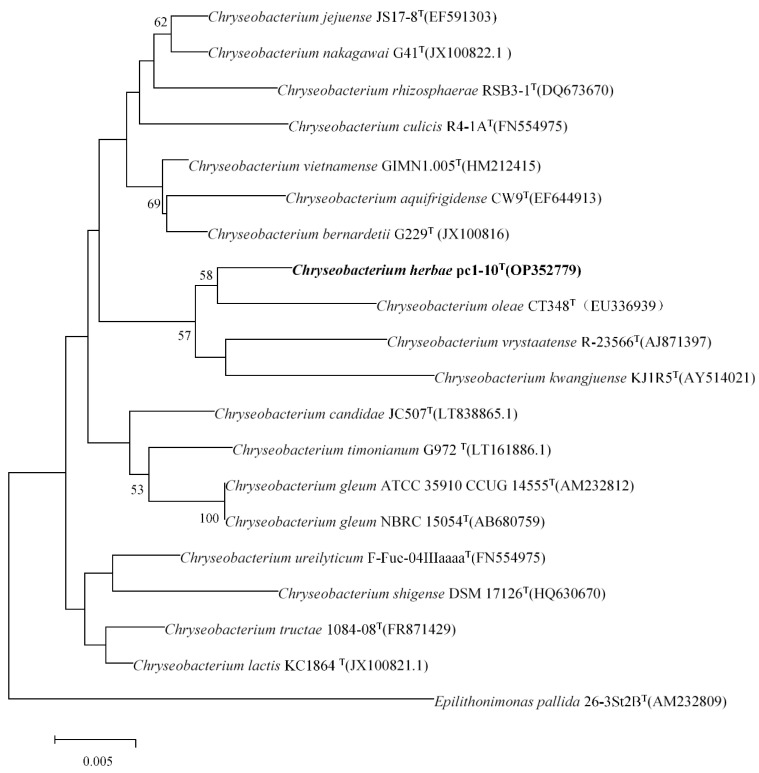
Neighbor-joining phylogenetic tree based on a comparison of the 16S rRNA gene sequences of strain *herbae* pc1-10^T^ and its closest relatives. GenBank accession numbers are given in parentheses. Bar, 0.005 nucleotide changes per 1000 nucleotides. Bootstrap values (>50%) based on 1000 replications are shown at branch nodes. Bar, 0.005 substitutions per nucleotide position.

**Figure 2 microorganisms-11-02017-f002:**
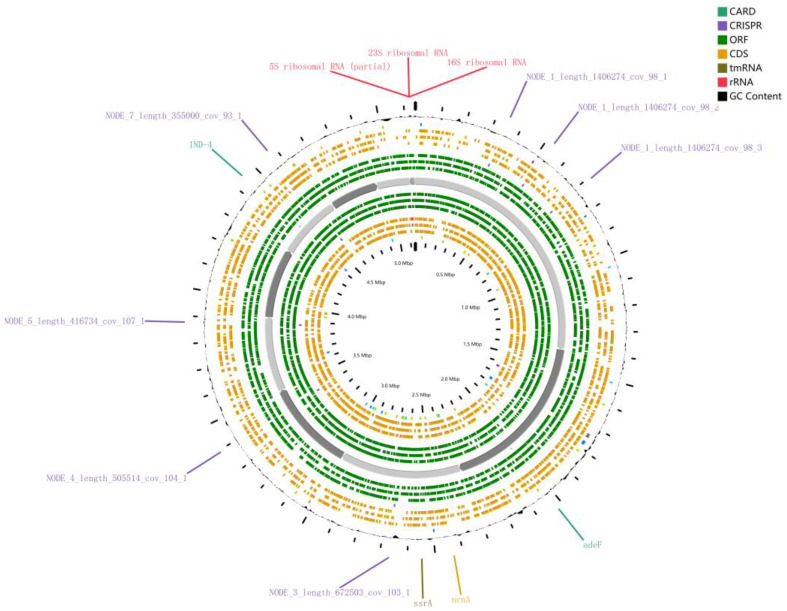
Annotation of the genome of *herbae* pc1-10^T^ through Proksee. CARD: The comprehensive antibiotic resistance database, CRISPER: CRISPR arrays, ORF: Open reading frames, CDS: Coding sequence, tmRNA: Transfer-messenger RNA, rRNA: Ribosomal RNA.

**Figure 3 microorganisms-11-02017-f003:**
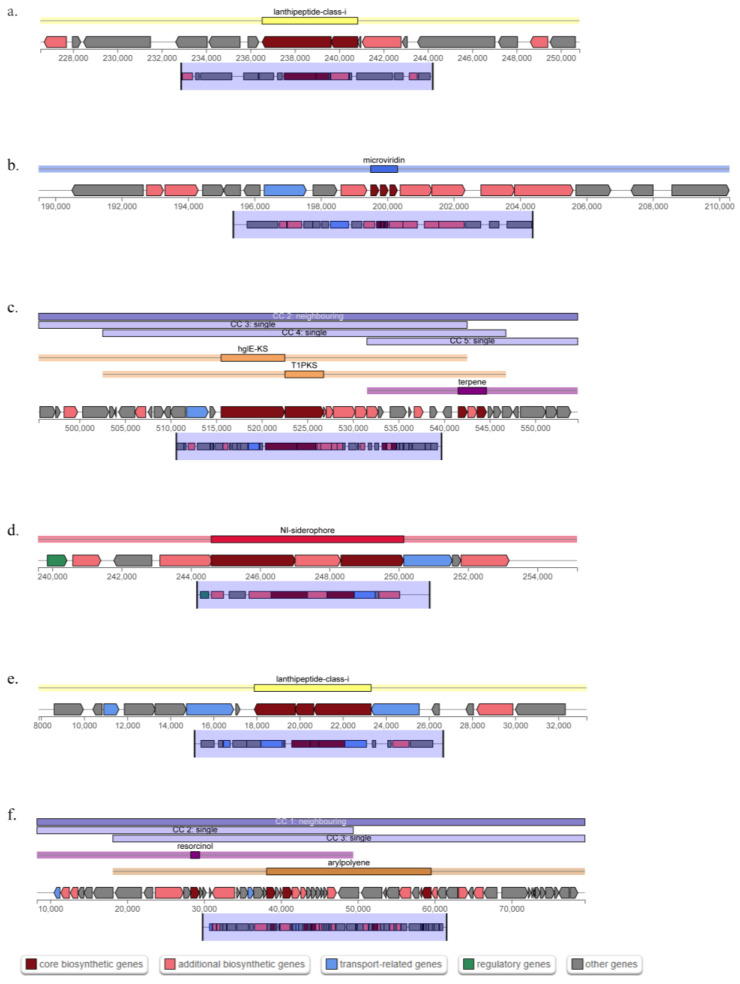
Biosynthetic gene clusters in the genome of *Chryseobacterium herbae* pc1-10^T^ retrieved using antiSMASH version 7.0.0. (**a**) Lanthipeptide class i. (**b**) Microviridin. (**c**) hglE-KS, T1PKS, terpene. (**d**) NI-siderophore. (**e**) Lanthipeptide class i. (**f**) Arylpolyene and resorcinol.

**Table 1 microorganisms-11-02017-t001:** Cellular fatty acid contents of strain *herbae* pc1-10^T^ and its relative reference strains in the genus *Chryseobacterium*. Strains: 1, *C. herbae* pc1-10^T^; 2, *C. oleae* CT348^T^; 3, *C. kwangjuense* KJ1R5^T^; 4, *C. vrystaatense* R-23566^T^. The data of other strains are from previous studies. Values were percentages of total fatty acids. Tr, Trace amount (<0.5%); ND, not detected; NA, not available.

	Percentage			
Fatty Acids	1	2 ^a^	3 ^b^	4 ^c^
C_16:0_	1.16	3.3	1.2	1.1 ± 0.3
C_16:0_ 3-OH	ND	0.5	Tr	1.3 ± 0.3
iso-C_13:0_	1.63	Tr	1.7	1.1 ± 0.4
iso-C_15:0_	42.82	38.5	45.4	41.8 ± 1.4
iso-C_15:0_ 3-OH	3.79	1.6	4.2	2.7 ± 0.3
iso-C_16:0_	0.82	3.9	ND	ND
iso-C_16:0_ 3-OH	1.18	2.3	ND	Tr
iso-C_17:0_	0.71	1.3	1.4	Tr
iso-C_17:0_ 3-OH	21.03	8.7	10.8	15.4 ± 1.8
iso-C_17:1_ ω9c	ND	ND	11.6	19.7 ± 2.3
anteiso-C_15:0_	1.24	4.8	Tr	1.7 ± 0.7
Summed features 3	8.3	14.7	15.4	9.1 ± 0.9
Summed features 9	12.43	14.8	ND	NA

^a^, Data from del Carmen Montero-Calasanz, Maria, et al. [56]. ^b^, Data from Sang, Mee Kyung, et al. [57]. ^c^, Data from De Beer, Hanli, et al. [58].

**Table 2 microorganisms-11-02017-t002:** Differential characteristics of strain *herbae* pc1-10^T^ and its relative reference strain in the genus *Chryseobacterium*. Strains: 1, *C. herbae* pc1-10^T^; 2, *C. oleae* CT348^T^; 3, *C. kwangjuense* KJ1R5^T^; 4, *C. vrystaatense* R-23566^T^. The data of other strains are from previous studies. Symbols: +, positive; −, negative; w, weak reaction. nm, not mentioned.

Characteristic	1	2 ^a^	3 ^b^	4 ^c^
Color of colonies	orange	yellow	yellow	yellow
Gram-stain	−	−	−	−
Growth temperature (°C)	10–30	5–35	10–38	4–32
Optimal growth (°C)	25	−	28–38	5
pH range for growth	5.0–9.0	5.0–8.0	6.0–8.0	nm
Optimal	6.0		7.0–8.0	nm
Growth in NaCl (%*w*/*v*)	0–1.8	0–1	1–3	1–2
Oxidase	−	+	nm	+
Catalase	+	+	nm	+
Motility	−	−	−	−
Esculin	−	+	+	+
Starch	+	+	+	−
Tween 40	+	+	nm	+
Tween 80	+	nm	+	+
β-Galactosidase	−	−	−	nm
Nitrate reduction	−	−		−
Indole production	+	+	+	nm
D-Glucose fermentation	−	+	nm	+
Urease	−	−	−	+
Mannitol	w	+	+	nm
D-Maltose	−	+	+	nm
Mannose	w	+	nm	+
DNA G+C content (mol%) *	37.9	38.2	40.2	37.1

* DNA G+C content (mol%) was obtained from genomic data. ^a^, Data from del Carmen Montero-Calasanz, Maria, et al. [56]. ^b^, Data from Sang, Mee Kyung, et al. [57]. ^c^, Data from De Beer, Hanli, et al. [58].

**Table 3 microorganisms-11-02017-t003:** Analysis of secondary metabolite biosynthesis gene clusters of *herbae* pc1-10^T^ through antiSMASH.

Region	Type	From-To	Most Similar Known Cluster		Similarity
Region 1.1	lanthipeptide-class-i	226,529 250,834			
Region 2.1	microviridin	189,501 210,305			
Region 2.2	hglE-KS, T1PKS, terpene	495,527 554,653			
Region 6.1	NI-siderophore	239,589 255,146	fulvivirgamide A2/fulvivirgamide B2/fulvivirgamide B3/fulvivirgamide B4	Other	33%
Region 7.1	lanthipeptide-class-i	7895 33,311	pinensins	RiPP	17%
Region 8.1	resorcinol, arylpolyene	8270 79,537	flexirubin	Polyketide	75%

## Data Availability

The datasets generated during and/or analyzed during the current study are available from the corresponding author on reasonable request.

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
