# Peer review of "Chryseobacterium herbae* Isolated from the Rhizospheric Soil of *Pyrola calliantha H. Andres* in Segrila Mountain on the Tibetan Plateau"

_microorganisms, 2023, doi:10.3390/microorganisms11082017_

Round 1

Reviewer 1 Report

Comments to „Chryseobacterium herbae isolated from the rhizospheric soil of Pyrola calliantha H. Andres. in Segrila Mountain on the Tibetan plateau“, submitted by Zhang et al.

 The interesting manuscript presents the identification of a Chryseobacterium herbae strain existing in the rhizosphere of an alpine wintergreen species. The authors emphasize on the phylogenetic and genotypic analysis, which is well done by the use of common, adequate methods. Also the genome analyses are convincing and worthwhile to get published.  Zhang et al. include some „Physiological, biochemical, and chemotaxonomic analysis“ in addition. This part gives some information, but the used methods are all insuffcient and rather old. They are not state of the art. Regarding lipids and and secondary metabolites, including toxic and modifies polypeptides, no current MS-MS methods were used for analyses. The LC-MS method for isoprenoid quinones is from 1984. Thus, the results might be incomplete, which is surely the case with the lipids. If there is the possibility, the authors should repeat and complete at least the identification of secondary metabolites, because they are of crucial importance.

Unfortunately, proteomics were not done. Therefore it is unknown whether the genes/gene clusters involved in secondary metabolite biosynthesis are active or not. Are bacterial small molecule secondary metabolites released into the culture medium? Also,  it remains unknown, if the beta-lactamase or/and a multidrug efflux transporter is/are active or not. This question is directly related to the habitat of the bacterium, as the rhizosphere should be enriched with (released) secondary metabolites of the plant. Pyrula species are known for their biocidal secondary metabolites, which could be fatal to microorganisms in the neighborhood of the roots. Regarding the ecological aspects, which belong to a characterization, the manuscript is unfortunately  incomplete. The manuscript would be greatly improved by identifying the released bacterial secondary metabolites.

Dear Editors,

the manuscript is nice but could be greatly improved by presenting released secondary metabolites.

Reviewer 2 Report

The manuscript represents the description of a new species belonging to the genus Chryseobacterium. The physiological, biochemical and genetic information about the newly described species and its closest relatives is given. There are some points that should be addressed by the authors.

 First of all, the manuscript requires English editing. The text is comprehensible but it may become easier to understand.

L. 17: The word “type” should be removed.

L. 25: When the genus name of a group is mentioned for the first time, it should be a full name (Chryseobacterium). Later in the text of the abstract the first letter only may be used (C.).

L. 21: “After simple functional verification” may be omitted in the abstract.

L. 30: Chryseobacterium herbae sp. nov.  is proposed…

  L. 42-43: Scarce or absent? If there is any, please give reference(s).

 L. 58-62: the list of the environments inhabited by members of Cryseobacterium genus should be arranged more properly. Now it looks like a random mixture of habitats.

Probably the Introduction section would look better if it starts with the description of genus Chryseobacterium followed by the description of geographical position and plant characterization.

 L. 64: “to characterize a new species …”

 L. 68: The strain of bacteria Chryseobacterium sp. pc1-10…

 L. 69: Pyrola rotundifolia

 L. 75-76: if you do not mention the taxonomic position and any other information about all the 15 colonies further in the text, it would be better to omit “of different in total, belonging to 8 genus and 4 phylums”.

 L. 76, 81, 102, 115, 139, 147, 156, 168, 185…: is the strain name “herbae pc1-10“, “pc1-10“ or “herbae pc1-10”? Please use one name throughout all the manuscript.

 L. 83-87 and 88-92 seem to duplicate each other. Please check.

 L. 97: did you mean “temperature gradients” of “temperatures”?

 L. 100: It would be better to use “optimum” instead of “ideal”.

 L. 102: “inoculated” instead of “infected”.

 L. 103: “…in order to determine the production of…”

 L. 104: “The culture was incubated for 48 hours…”

 L. 133: “Well-grown cells of strain … grown on … were used for fatty acid analysis”.

 L. 162: Strains

 L. 177: “yet tolerating”

 L. 240-270: the bare enumeration of the number of genes associated with various categories found with RAST does not really make sense. It may be more valuable to determine the differences in gene presence or gene cluster organization between the newly described strain and its closest relatives, especially if it coincides with physiology, as it is done in the following abstracts. Fig. 2, therefore, may be omitted or sent to Supplementary files.

 L. 311: “Cells of a new strain herbae pc1-10T…”

 L. 316: “…are hydrolysed but arabinose or urea are not decomposed.

 At the end of an article describing a new taxon there should be a short section “Description of Chryseobacterium herbae sp. nov.” consisting of a brief summary of a description. Please find examples at the International Journal of Systematic and Evolutionary Microbiology and follow them.

The manuscript requires English editing. The text is comprehensible but it may become easier to understand.

Round 2

Reviewer 2 Report

The authors have made revisions to the original text but there is still some confusion left. Please carefully check the text.

L. 17: “bacteria” is plural, so please change all the sentence to accomodate either singular or plural.

L. 21-22: The sentence should be rephrased.

L. 26-27: The period after the word Chryseobacterium must be removed.

L. 32: “sp. nov.” should not be italicized.

L. 49: Animals and plants are a sort of environment as well. Please use other word(s) describing the references 24-29.

L. 177: What do you mean by “lower cells”? Is it a pellet?

L. 182: What tense is the sentence used in? Is it Present or Past? Please make corrections.

L. 257: “The data of other strains are from previous studies.” Please include references to these studies.

L. 303: The sentence starting with “Preliminary demonstration of…” lacks a verb and it is not clear what you wanted to say.

L. 306, Fig. 2: Although a brief description of the findings with RAST annotation was added instead of enumeration of genes, the pie diagram and data for subsystem categories still seem inappropriate without the comparison with the relatives or organisms with similar functions. Thus I would still strongly recommend sending it to Supplementary files.

P. 377-378: It would be better to use “optimum” instead of “ideal”.

P. 381: What produces gelatinase? There is no noun in the sentence.

P. 420: “Due to the fact…”: the sentence seems to be unfinished.

After a good work done there is still some work to do. Incomplete sentences are of most concern. Please carefully check the text.
